# Validity of the Stryd Power Meter in Measuring Running Parameters at Submaximal Speeds

**DOI:** 10.3390/sports8070103

**Published:** 2020-07-20

**Authors:** Frank Imbach, Robin Candau, Romain Chailan, Stephane Perrey

**Affiliations:** 1Seenovate, 34000 Montpellier, France; romain.chailan@seenovate.com; 2INRAe–DMeM, Univ Montpellier, 34000 Montpellier, France; robin.candau@umontpellier.fr; 3EuroMov Digital Health in Motion, Univ Montpellier, IMT Mines Ales, 34090 Montpellier, France; stephane.perrey@umontpellier.fr

**Keywords:** validity, running power meter, force platform, leg stiffness, Bayesian analysis

## Abstract

This study assessed the Stryd running power meter validity at sub-maximal speeds (8 to 19 km/h). Six recreational runners performed an incremental indoor running test. Power output (PO), ground contact time (GCT) and leg spring stiffness (LSS) were compared to reference measures recorded by portable metabolic analyser, force platforms and motion capture system. A Bayesian framework was conducted for systems validity and comparisons. We observed strong and positive linear relationships between Stryd PO and oxygen consumption ($R^{2}=0.82$, $BF_{10}>100$), and between Stryd PO and external mechanical power ($R^{2}=0.88$, $BF_{10}>100$). Stryd power meter underestimated PO ($BF_{10}>100$) whereas GCT and LSS values did not show any significant differences with the reference measures ($BF_{10}=0.008$, $BF_{10}=0.007$, respectively). We conclude that the Stryd power meter provides valid measures of GCT and LSS but underestimates the absolute values of PO.

## 1. Introduction

The last decade was marked by technological improvements in wearable sport devices to quantify the exercise features in ecological conditions. Endurance runners showed a large interest for non-differential global positioning systems (GPS), stride sensors and heart rate monitors integrated in sport watches. Such devices allow them to quantify their exercise and to program their training protocols. Parameters of importance are volume and intensity, commonly estimated by physical measures (e.g., distance covered, elevation, speed) and physiological markers (i.e. indexes mostly assessed from exercise heart rate). In addition to its own sensations, all recorded data allow the runner to adapt its pace in real time as well as being used to quantify the training loads [1]. Despite some validity and reliability studies carried out in laboratory at sub-maximal speeds [2,3], GPS measures are unsuitable in indoor conditions. Used outdoor, the GPS signal may be altered by atmospheric conditions and local obstructions such as forest, sharp mountains or city buildings [4]. Besides, low frequency, sharp turns, high speeds and direction changes that are found in running cause measurement errors [5,6]. Compared to accelerometer measures, the energy expenditure estimate from GPS watches seems to be low reliable [7] and its validity remains exercise intensity dependent [8]. Finally, without precise measures of elevation, hill running exercise challenges their usefulness to the extent that speed estimate is no longer a suitable intensity training parameter as determinant of the performance. In this context, power meters express their interest, reinforcing the use of heart rate monitors and surpassing the limits of GPS measurements by providing an alternative and transversal measure of exercise intensity.

Appeared first in cycling [9], power meters using a strain gauge instrumented pedals system [10] revolutionized training and performance assessment [11]. Such a measure enabled the cyclist to model its own record power profile, as a performance signature [12] and to estimate useful physiological parameters for training programming [13].

Running power meters, including inertial measurement unit (IMU), emerged the last five years as foot pods or hip positioned IMU. Aiming to estimate metabolic demand, mechanical work, running gait and stride characteristics, they could be useful in a performance optimization and in an injury prevention context on all types of terrain [14,15]. However, as for the former devices, scientific assessments of their reliability and validity are required. Otherwise, cautions should be taken when interpreting the measurements.

The Stryd power meter (Boulder, Colorado, United States) is a pioneer in the field and provides the following measures in real time: pace, running power output (PO),vertical oscillation, elevation, distance, ground contact time (GCT) and leg spring stiffness (LSS). As a novelty among wearable sensors, only a few recent studies assessed its validity and reliability. Contact time compared to 3D motion analysis is prone to loose accuracy with an increase of speed [16]. Pace (and derived total number of steps) has been shown valid and reliable during hiking and trail running bouts [17]. Furthermore, PO was reliably measured on different surfaces at sub-maximal speed corresponding to 85% of the individual lactate threshold [18]. The PO-velocity relationship was highly linear when tested on a treadmill for a 8 km/h to 20 km/h velocity range [19]. However, as the algorithms used to derive the metrics from the Stryd sensor are proprietary, the estimate of these metrics remains unclear. To our knowledge, these aforementioned studies compared neither PO estimates, nor LSS estimates to reference systems. Thus, their absolute and relative validity still has to be explored.

Through IMU technology and based on the previous findings, we hypothesised that the Stryd system provides valid measures of stride parameters (PO, LSS and GCT). Accurate measures would allow the runner to get real time precious information about exercise intensity and muscle fatigue. Thereby, such measures could be requisites for a performance optimisation by programming appropriate training sessions and by monitoring running performances over a season. Recreational runners, practising for health, could also profit from these measures warning about stride impairments, usually responsible for injuries and mostly influenced by running exposure weighted by individual properties (e.g., body mass index, age etc.) [20].

The aim of our study was to assess the validity of the PO, LSS and GCT measures from the Stryd footpod at different sub-maximal running speeds by comparing them to reference systems in ecological conditions. To do so, validated and reference methods were used to calculate the external mechanical power and the leg stiffness from accurate force platform measures [21,22].

## 2. Materials and Methods

### 2.1. Experimental Approach

To assess the validity of the Stryd power meter, we compared the recorded data with force plates, motion analysis system and a portable metabolic analyser on a 200 m indoor running track.

### 2.2. Participants

Six recreational runners (four male age: 39 ± 3 years, $\dot{V}O_{2}\max:53.85\pm 6.09\;{\mathrm{mlO}}_{2}$·$\min^{-1}$·${\mathrm{kg}}^{-1}$ and two females age: 35, $\dot{V}O_{2}\max:48.33\pm 6.75\;{\mathrm{mlO}}_{2}\cdot \min^{-1}$·${\mathrm{kg}}^{-1}$) voluntary participated in this study. All participants validated the inclusion criteria: (i) older than 18 years; (ii) train 3 to 5 times a week and on a treadmill at least once per week; (iii) not suffering from any injury impacting running capacities for the last 6 months. The study was performed in agreement with the standards set by the declaration of Helsinki (2013) involving human subjects. Following an explanation of all procedures, risks and benefits associated with the experimental protocol, each participant gave his written informed consent prior to experimentation. The protocol was reviewed and approved by the local research Ethics Committee (IRB-EM 1901-B, EuroMov, France). Informed written consent was obtained before the experimental testing sessions.

### 2.3. Protocol

The study was conducted in April 2019 when participants were preparing themselves for a long distance race (over 42 km). The test consisted of an incremental running trial around a 200 m track (Figure 1). The initial speed was set at 10 km/h and 8 km/h for men and women, respectively. Thereafter, the speed was increased by 0.5 km/h every minute. Cones were set at 20 m intervals along the 200 m track (inside the first line). The running pace was dictated by audio signal and the runners had to be within 2 m of the cones at each beep signal. When a runner was behind a cone for three consecutive times, the test was stopped. Individual maximal aerobic speed (MAS) was determined as the lowest speed at which $\dot{V}O_{2}\max$ was attained [23]. MAS was reached between 12 and 20 min of exercise in order to limit impairments caused by the accumulation of fatigue (allowing MAS values to reach approximately 20 km/h at 20 min). The participants were rested before the start, the first minutes of the test acting as a warm-up.

### 2.4. Materials

#### 2.4.1. Power Meter

Each participant wore the Stryd power meter, a foot-mounted inertial sensor of 9.1 g reinforced with carbon fiber, firmly attached on the shoe and according to manufacturer recommendations. The device stores at 1 Hz sampling frequency the following variables: GCT, vertical oscillation, running PO, distance, LSS, cadence. According to Stryd team information, the device is operational out of the box and should not need any calibration, accepting a measurement error of 3 percent. Participants filled in their body height and body mass prior its use, requisites for the PO estimation. As a precautionary measure, the device was fully loaded and activated 20 min before the beginning of the test. The firmware version used was the 1.1.9 (released on February 2019) and data were extracted in flexible and interoperable data transfer (FIT) format from the Stryd application (http//www.stryd.com/powercenter). To analyse data extracted from the Stryd power meter, we converted FIT files to comma-separated values format by parsing files with Python (version 3.7, “fitparse” library).

#### 2.4.2. Gas Exchange Measures

Participants wore the Cosmed k4b2 portable metabolic system in order to record breath-by-breath gas exchange measures. The device has been validated by several independent authors [24,25,26]. The device was checked and certified valid by the Cosmed company two months prior to the study. Before each testing sessions, the metabolic analyser was powered on to warm up for 30 min and then ${\mathrm{CO}}_{2}$ and $O_{2}$ sensors were calibrated based on known gas tank concentrations and ambient air measurements. Flow meter calibration was then completed using a 3.0 L syringe according to the manufacturer recommendation. Calibration was done for each subject after 30 min warm-up activation. To produce uniform sampling for subsequent numerical analysis, we linearly interpolated data on a second-by-second basis. Due to a noisy signal provided by the portable metabolic system, data were averaged with a fifth order moving average filter [27], corresponding to 5 s time bins [28].

#### 2.4.3. Force Platforms

A track embedded force platforms system measured the ground reaction forces (GRFs) once per lap (200m) during the incremental running test. The system consisted of three force platforms (one Kistler, Switzerland and two AMTI, USA) of dimensions 90 cm * 90 cm connected in series and covered with a tartan mat. Each platform was calibrated before the study. Sampling frequency was set at 500 Hz.

#### 2.4.4. Motion Analysis

The entire runner stride measurements along the platform section were performed using the Coda Motion 3-D movement analysis system (Charnwood Dynamics Ltd., United Kingdom). The system was composed of marker devices, sensor modules and data analysis software. The marker devices consist of infrared emission markers. The sensor module is made up of three optical sensors, which capture 3-D position and orientation by tracking the markers in real-time. Two sensor modules were placed on both sides of the platform area and two markers (CXS models) were firmly fixed to the heel and on the fifth metatarsal bone of the runner. The system delivers reliable real-time 3D measurements on the computer screen throughout the experiment with a 400 Hz sample frequency. The data were processed with the Coda Motion Odin software platform.

### 2.5. Calculations

#### 2.5.1. Ground Contact Time, Stride Time and Stride Frequency

Changes in vertical GRFs signal were used to detect foot strike and to estimate GCT (ms), defined by the duration of the GRFs variation. GCT (ms) was defined as the duration of the vertical GRFs signal. Flight time was estimated from the Z axis heel marker changes monitored by the Coda Motion analysis system. Assuming that participants have an equal stride properties between lower limbs, the stride time was approximated as following:$$Ts=\frac{C_{2}-C_{1}}{2},$$
with Ts the stride time in seconds, $C_{2}$ and $C_{1}$ respectively the time instances (s) of the second and first heel strikes recorded by the optical sensor located on the foot. Hence,
$$\omega =\frac{1}{Ts}$$
is the stride frequency (Hz).

#### 2.5.2. External Mechanical Power, Mechanical Cost of Running and Mechanical Efficiency

We calculated the external power and the mechanical cost of running following an external energy summation approach. GRFs signals were computed in the anterior-posterior (x) and vertical (z) axis. We omitted the lateral axis due to its negligible contribution when running on a track [21]. First, the acceleration $\vec{A}$ (${m/s}^{2}$) is decomposed on both *z* and *x* axes, respectively defined as
$$\begin{array}{cc} Az & =\frac{\left(Fz-mg\right)}{m}\;\mathrm{and}\\ Ax & =\frac{Fx}{m}, \end{array}$$
where Fz and Fx are the components of the force recorded (N), *m* is the mass of the subject (kg) and g=9.80665 is the acceleration due to the gravity (${m/s}^{2}$). Consequently, the speed at time *i* denoted ${\vec{S}}_{i}$ (m/s) is
$$\begin{array}{cc} {\vec{S}}_{i} & =\int_{t=i-1}^{i}{\vec{A}}_{t}\;dt+{\vec{S}}_{i-1}\;\mathrm{and}\\ {\vec{D}}_{i} & =\int_{t=i-1}^{i}{\vec{S}}_{t}\;dt+{\vec{D}}_{i-1}, \end{array}$$
with ${\vec{D}}_{i}$ the distance (m) at time *i* and *t* the time measurement according to the sampling frequency (2 ms). Potential, kinetic and total work ($W_{p}$, $W_{k}$ and $W_{t}$) were calculated as
$$\begin{array}{cc} W_{p} & =mg\left(\max D_{z}-\min D_{z}\right)\\ W_{k} & =\frac{1}{2}m\left(\max S_{x}^{2}-\min S_{x}^{2}\right)\\ W_{t} & =W_{p}+W_{k}, \end{array}$$
where *z* and *x* are the vertical and anterior-posterior axes, respectively. Finally, external mechanical power (W) was calculated by
$${\dot{W}}_{ext}=W_{t}\;\omega .$$

The mechanical cost of running (Cm) in J · ${\mathrm{kg}}^{-1}$·$m^{-1}$ was thus defined as
(1)$$Cm={\dot{W}}_{ext}\;S_{x}^{-1}\;m^{-1}.$$

Metabolic power (${\dot{W}}_{met},W$) and net Mechanical Efficiency (ME, %) were estimated from $\dot{V}O_{2}$ measurements using energy equivalent of $O_{2}$ as following
$${\dot{W}}_{met}=\frac{\dot{V}O_{2}\;m}{60}\;k$$
(2)$$ME=\frac{{\dot{W}}_{ext}}{\Delta {\dot{W}}_{met}}\;100,$$
with $\dot{V}O_{2}$ in ${\mathrm{mlO}}_{2}$·${\mathrm{kg}}^{-1}$·$\min^{-1}$, *k* the energy equivalent for the consumption of $1\;{\mathrm{mlO}}_{2}$ for a value of 21.1J [29,30] and $\Delta {\dot{W}}_{met}$ the variation of $\dot{V}O_{2}$ above resting.

#### 2.5.3. Leg Stiffness

Stryd LSS metric was compared with the reference method [22] for assessing leg stiffness ($k_{leg}$, kN·$m^{-1}$), using force platform measures and defined as
(3)$$k_{leg}={\hat{F}}_{z}\;\Delta L^{-1},$$
where ${\hat{F}}_{z}$ denotes the maximal values of $F_{z}$ and ΔL=Δy+L(1−cosθ). In the latter, $\theta =\sin(vT_{c}/2L)$, Δy is the vertical displacement of the center of mass, *v* is the forward velocity, $T_{c}$ is the contact time and *L* is the initial leg length standardized at 0.53 of the subject’s height.

#### 2.5.4. Time Matching

The following procedure was used to match power meter and force platform measurements. After checking the length of the track (200 m following the inner line), participants started to run right after the force platforms set up, spreading out over 9 m (Figure 1). The recorded sequences were matched by subtracting 9 m to each 200 m estimated by the power meter, assuming that the Stryd device measures distance reliably [31]. To keep one value for each metric of interest, values were averaged over this distance and over the three force platforms.

### 2.6. StatisticaL Analysis

Statistical modelling in a small data set context can lead to statistical power issues and may suffer from biased parameters estimation. To tackle this issue, the modelling was conduced in a Bayesian framework. We counterbalanced the lack of data (participants) by providing a priori information inside the models, based on empirical knowledge and literature. The Hamilton Monte Carlo algorithm was used to infer the parameters of models and caution has been taken to diagnose their convergence [32]. To figure out the relevance of variable inclusion in the models and to provide an alternative to significant testing of the null hypothesis ($H_{0}$), we computed Bayes Factors ($BF_{10}$). Such a factor represents a continuous measure of evidence for the alternative hypothesis ($H_{1}$) over $H_{0}$. Based on theory of Jeffreys [33] and according to guidelines of Lee and Wagenmakers [34], we provided the following classification for interpretation: $BF_{10}⩾$ 100, 30–100, 10–30, 3–10, 1–3 correspond to an extreme, very strong, strong, moderate and anecdotal evidence for $H_{1}$, respectively. A $BF_{10}$ of 1 means there is no evidence of an hypothesis over the other. Below this value, the evidence is against $H_{1}$ or for $H_{0}$ following the inverse of the mentioned scale.

#### 2.6.1. Reference Measures

To ensure that reference measures were valid, a Linear Mixed Model (LMM) was computed to evaluate the relationship between the Cm (Equation (Equation 1)) and speed calculated from force platform measurements. Speed and participants were settled as fixed and random effects respectively, in order to consider the variability of Cm among participants. Relationship between the portable metabolic system-derived variables and the force platform-derived variables were also assessed by computing a LMM where the mechanical power and participants corresponded to fixed and random effects, respectively. For each LMM, an intraclass correlation coefficient (ICC) was reported to highlight the fraction of the total variance in the data accounted for between-subject variation. It justifies the inclusion of participants as random effects in the model. Finally, linear models were used to examine the mean relationships between (i) GCT and running mechanics (running speed and frequency), (ii) leg stiffness and running mechanics. For the sake of clarity, those models are detailed in supplementary materials (Appendix A). Coefficient of determination and 95% credible intervals (CI) were reported in order to quantify the degree of linear relationship.

#### 2.6.2. Mechanical Power

To assess the linear relationship between the Stryd mechanical power and the metabolic energy consumption during the MAS test, we defined the following model,
(4)$$y_{i}=\beta_{0}+\beta_{1}P_{mec}+\epsilon_{i},$$
with $y_{i}$ being the $\dot{V}O_{2}$, $\beta_{0}$ denoting the intercept, $\beta_{1}∼N\left(0,10\right)$ being a weakly informative prior and $\epsilon_{i}$ the error term.

In addition to coefficient of determination and 95% CI, correlation between variables were observed through pair-wise Bayesian correlation tests using non-informative Jeffrey’s priors [33]. A similar linear model (Equation (Equation 4)) was computed to assess the relationship between PO measured by the force platform and PO assessed from the power meter.

Thereafter, the effects of the system of measurement (force platform and Stryd power meter), the speed and the participant were evaluated. Due to random measurement errors and technological issues (e.g., mismatch between foot strikes and the force platform area), it has been required to deal with missing data as well as different number of repeats between systems. Consequently, a LMM was preferred to a repeated measures analysis of variance. Through a design-driven approach [35], subject varying intercept and slopes were included in addition to fixed effects. This model allowed us to consider the inter-subject and intra-subject variability (e.g., heterogeneous PO levels and individual PO kinetics in response to speed and strides changes). Independent variables were standardised prior modelling, easing the interpretation by allowing the direct comparisons of estimated parameters. The model was defined as,
(5)$$\begin{array}{cc} y_{ijk}=\beta_{0} & +S_{0i}+\beta_{1}\;device_{k}\\ \; & +\left(\beta_{2}+S_{1i}\right)\;speed_{j}\\ \; & +\beta_{3}\left(speed_{j}\;device_{k}\right)+\epsilon_{ijk}\;, \end{array}$$
with $y_{ijk}$ being the response variable for a subject *i*, speed *j* and device respectively *k*. In this equation, $\beta_{0}$ denotes the fixed effect intercept, $S_{0i}$ is the offset term intercept which represents the deviation from $\beta_{0}$ for the subject *i*, $\beta_{n}$ are the parameters for each corresponding predictor, $S_{1i}$ is the random slope for each subject and $\epsilon_{ijk}$ is the observation-level error. Priors were chosen according to empirical knowledge and literature. Because the relationship between devices (Stryd power meter and force plateforms) remains unknown but presuming an underestimate of the Stryd power meter, a vague prior was fixed such as $\beta_{0}∼T\left(3,0,10\right)$ and $\beta_{1}∼N\left(0,1000\right)$. According to the well-known strong and positive relationship between running speed and external mechanical power [36], a weakly but more informative prior (i.e. with a lower variance) was fixed to the speed parameter with $\beta_{2}∼N\left(0,200\right)$.

#### 2.6.3. Ground Contact Time

In the same way, effects of the MAS test parameters on GCT were assessed through the Bayesian LMM (Equation (Equation 5)). Weakly informative priors were assigned to the parameters of the force platforms, power meter and speed. This choice was motivated from empirical knowledge and previous findings about changes in contact time with velocity [37]: $\beta_{0}∼T\left(3,0,10\right)$, $\beta_{1}∼N\left(0,1\right)$, $\beta_{2}∼N\left(0,1\right)$.

#### 2.6.4. Leg Stiffness

The last variable of interest was analysed following the same procedure (Equation (Equation 5)). Weakly informative priors were chosen according to the trust in leg stiffness values [38] in order to estimate the posterior distributions for each parameters as $\beta_{0}∼T\left(3,0,10\right)$, $\beta_{1}∼N\left(0,10\right)$, $\beta_{2}∼N\left(0,1\right)$.

Bayesian models were computed in the probabilistic Stan programming language. Bayes factors were estimated through a bridge sampling and using the R package “brms” [39]. Finally, agreement between the power meter and force platforms measures was described by a Bland-Altman analysis for each variable of interest [40]. All statistical analysis were computed with R software (version 3.5.3).

## 3. Results

### 3.1. Reference Systems: Force Platforms, Portable Metabolic System and Motion Capture

#### 3.1.1. Mechanical Cost of Running

Using the force platforms, we calculated the mechanical cost of running following the Equation (Equation 1). Mean values of Cm were 2.36±0.46J·${\mathrm{kg}}^{-1}$·$m^{-1}$. By comparing the Cm with the increase of speed, we observed a moderate negative linear relationship for all participants ($R^{2}=0.66$, [0.60,0.71]95%CI). Both speed (as a component of Cm calculation) and subject have shown an effect on the Cm measure with an extreme evidence ($BF_{10}>100$). In terms of explained variance by participant effect, ICC reported a moderate correlation (ICC=0.65, [0.34,0.94]95%CI)) supporting the individual differences in Cm. Highest values of Cm were found at low speeds (up to 3 m/s).

#### 3.1.2. Metabolic and External Mechanical Power Relationship

Consumed metabolic energy ($\dot{V}O_{2}$) and external mechanical power revealed a strong and positive linear relationship for the 6 participants ($R^{2}=0.85$, [0.76,0.89]95%CI). Results of the LMM (Table 1) showed a significant effect of mechanical power over $\dot{V}O_{2}$ ($\beta_{1}=0.081$, [0.035,0.111]95%CI). Bayes factor supported the results with an extreme evidence of both external mechanical power and subject effect ($BF_{10}>100$). Large standard deviations of intercept parameter indicated heterogeneous levels between participants. ICC supported the individual differences with a high correlation (ICC=0.94, [0.55,1]95%CI)). The net mechanical efficiency was calculated from the force platform measures following Equation (Equation 2). Group mean value of ME was equal to 55±3%.

#### 3.1.3. Ground Contact Time and Leg Stiffness

Force platform GCT and $k_{leg}$ values according to the method of McMahon and Cheng [22] (Equation (Equation 3)) are represented in Figure 2. GCT showed a strong and negative linear relationship with both speed ($R^{2}=0.96$, [0.86,0.98]95%CI) and frequency ($R^{2}=0.93$, [0.79,0.96]95%CI), recorded by the force platform and the motion capture system, respectively. Bayes factors reported a moderate evidence for the alternative hypothesis ($BF_{10}=3.07$ and $BF_{10}=4.02$). $k_{leg}$ showed a strong and positive linear relationship with the stride frequency ($R^{2}=0.82$, [0.42,0.89]95%CI, $BF_{10}=16.12$). In contrast, $k_{leg}$ did not significantly increase with speed ($R^{2}=0.40$, [0,0.64]95%CI). The resulting Bayes factor ($BF_{10}=0.22$) indicated an anecdotal evidence in favour of the null hypothesis.

### 3.2. Stryd and Reference Measures Comparisons

#### 3.2.1. Consumed Metabolic Energy and Power Output

Individual consumed metabolic energy and PO relationships are represented in Figure 3. We truncated the first part of the signal in order to remove the habituation period, where PO increases instantly and $\dot{V}O_{2}$ is shortly delayed. A strong and positive linear relationship between $\dot{V}O_{2}$ and PO for all participants was observed ($R^{2}=0.82$, [0.81,0.83]95%CI, $BF_{10}>100$). This results supported a valid relative measure of the Stryd PO from low speeds until MAS. Bayesian pair-wise correlation coefficient indicated a strong and positive correlation between both parameters for all participants (r=0.90, [0.89,0.92]95%CI, $BF_{10}>100$).

#### 3.2.2. Mechanical Power

A descriptive analysis of PO differences between both measurement systems indicated the greatest differences at highest speeds, suggesting a proportional error (normalized PO differences per participants varied from 38% to 60% between the two systems). By modelling the averaged PO across participants, a strong and linear relationship was observed over the MAS test (Figure 4a, $R^{2}=0.94$, [0.91,0.95]95%CI, $BF_{10}>100$). However, the two systems were different regarding their absolute measures. To fix this issue, the linear model fitted on the averaged values of each participant measures could provide a correction function. In this study the estimated function was of the form f(x)=ax+b, with a=173.837 and b=1.569 (Figure 4b). In addition, the Bland-Altman analysis provided a representation of such differences in absolute values and a proportional error which raised with mechanical power increase, and so speed (Figure 5a).

In addition to averaged power comparisons, LMM mean posterior distributions and CI confirmed the underestimate of the Stryd PO ($\beta_{1}=-305$, [−324,−286]95%CI, $BF_{10}>100$). Results are reported in Table 2. In agreement with the relationship of mechanical power and speed, the posterior distribution of the speed parameter also reported a positive effect on PO with a strong evidence ($BF_{10}=15.38$). Stryd measure and speed interaction reported a negative effect. It suggests that the power meter measure is not homogeneous while speed raises (from 8 km/h to approximately 19 km/h). By comparing models with and without the interaction term, Bayes factor confirmed this effect with an extreme evidence ($BF_{10}>100$). The ICC revealed a high correlation (ICC=0.79, [0.55,0.96]95%CI, estimated error = 0.11). Such a correlation indicated an important variance explained by the random intercept and slopes per subject, justifying the relevance of subject random terms in the present modelling.

#### 3.2.3. Ground Contact Time

Results of the Bayesian LMM (Table 2) reported a negative although quasi-null posterior estimate of the Stryd parameter ($\beta_{1}=-0.005$, [−0.009,−0.002]95%CI). It suggested that a small but negative effect of the Stryd device exists. However, Bayes factor reported an extreme evidence in favour of the null hypothesis ($BF_{10}=0.008$). Hence, no significant differences were found between the systems of measurement. Posterior distribution of the speed parameter also reported a negative effect on GCT in agreement with our precedent results (see Figure 2). Bayes factor supported the importance of this effect with an extreme evidence ($BF_{10}>100$). However, not any effect were observed for the Stryd power meter and speed interaction. Bayes factor confirmed this result with an anecdotal evidence against the alternative hypothesis ($BF_{10}=0.72$). Thus, GCT seemed to be measured with consistency across the whole range of speeds. In terms of by subject explained variance, the ICC reported a moderate correlation (ICC=0.55, [0.24,0.89]95%CI, estimated error = 0.18). Therefore, individual differences in GCT were lower than PO differences.

#### 3.2.4. Leg Spring Stiffness

For the last measure of interest, any difference were found between devices over the LSS measurement ($\beta_{1}=-0.602$, [−2.334,1.154]95%CI) as reported in Table 2. An extreme evidence against the alternative hypothesis supported this result ($BF_{10}=0.007$). The speed did not appear to impact the LSS measure which remained quite constant. Bayes factor supported this result with a very strong evidence against the alternative hypothesis ($BF_{10}=>0.012$). Finally, the posterior distribution of the LSS parameter indicated that LSS was measured with consistency across the range of speed. A very strong evidence against the alternative hypothesis supported the negligible effect in modelling ($BF_{10}=>0.020$). Nonetheless, by between subject variability showed a moderate correlation (ICC=0.65, [0.34,0.93]95%CI, estimated error = 0.16) revealing differences in individual baseline levels as well as responses to speed increment.

## 4. Discussion

The present study aimed to assess the validity of the Stryd power meter at sub-maximal speeds. In the first part of the experiment, absolute values obtained from reference systems (portable metabolic analyser, force platforms and motion capture) were compared to the literature. Then, comparisons of mechanical variables were made between the Stryd power meter and the gold standard systems of measurement.

### 4.1. Reference Measures

Mechanical cost values found for the six participants (reported in the first part of the results) were in agreement with expected values when the total work is calculated by assuming no energy transfer between potential and kinetic energies [41,42,43,44]. High values of Cm at low speeds reveal inefficient running patterns, as described previously [41,44,45].

Oxygen uptake and work rate relationship were largely studied and well described by Poole et al. [46], and Gaesser and Poole [47]. Our results agreed with the literature with a strong and positive relationship between consumed metabolic energy and mechanical power across the MAS test (Table 1). To end with metabolic and mechanical power measures, the net mechanical efficiency (ME=55±3%) also confirmed suitable values for a running task, as stated in the literature [42,48,49].

Linear relationship (Figure 2) between both GCT, speed and stride frequency were in agreement with the literature [38,45]. The leg stiffness calculated following McMahon’s method [22] also reported consistent results with previous author findings [38,50,51]. Based on these results, we considered that our reference measures were suitable for comparisons with the Stryd power meter.

### 4.2. Power Meter and Reference Measurements Comparisons

Consumed metabolic energy and PO estimated by the Stryd power meter indicated a satisfying positive linear relationship for each participant. Individual regression slope and intercept differed between participants, according to heterogeneous body characteristics included in PO calculation (mainly body mass) and running performance level. To date, only three studies assessed the metabolic demand and Stryd PO relationship [18,52,53]. The last found weak linear relationship between PO and $\dot{V}O_{2}$ but suffered of methodological flaws, highlighted by Snyder et al. [54]. Our results appear to be more consistent than Austin et al. [52] but we found lower correlations than Lara et al. [18] or Stryd own researches [55]. It is important to point out the methodology employed when analysing $VO_{2}$ from Cosmed k4b2 and Stryd power data. In the present study, the first part of the running exercise (less than one minute of exercise) was omitted due to a normal time delay for $\dot{V}O_{2}$ to increase while PO increases instantaneously [56]. Moreover, as the portable metabolic analyser provides a noisy signal [25], filtering the data appears to be essential prior any analysis. The fifth order moving averaged filter allowed analysis on smoothed data but still sensitive to the exercise conditions. We encourage such a process for $VO_{2}$ analysis during incremental running test or at least to report details about data processing.

Despite a strong linear relationship between the Stryd power meter and the mechanical power calculated from the force platform (Figure 4a), absolute values have shown major differences. Results found from the linear mixed model (Table 2) and the descriptive analysis of differences between the Stryd power meter and the force platforms converged, suggesting a proportional error of the Stryd PO estimate with the increase of speed (Figure 5a). The underestimate of the power meter measure could be accounted for the apparent ME used in the power calculation. Apparent efficiency represents the ratio between mechanical energy and metabolic energy and is modulated by elastic storage and recoil from the eccentric to concentric phase. It has been measured during running, walking and cycling [42,48,49,57,58]. Researchers found values up to 53% in running versus approximately 30% and 25% in walking and cycling respectively (we found similar values of 55%, as reported above). Running power meters are concerned with ME assessment but the way it has been integrated in the PO estimation could explain large differences in absolute PO values. Stryd team mentioned in their white papers a gross ME equal to 25% that can be approached by elite runners [55]. This ME value is about an half of the apparent ME values reported by aforementioned authors and the one we found. Such a difference could explain the underestimate of PO from the Stryd power meter when compared to the systems of reference. To tackle this absolute error, we proposed a linear correction function adjusting the power value to the appropriate scale (Figure 4b). The provided function is estimated from data of only 6 runners. Even though their heterogeneous aptitude and body mass varied, further studies including more participants could provide a more accurate PO correction for a wide range of runners.

According to García-Pinillos et al. [16], an underestimate and a poor reliability for the GCT measure by comparing the power meter to an OptoGait system were found. No such results were found in the present study although an underestimate but negligible GCT measure was observed when compared to force platforms (Table 2, Figure 5b). A potential difference could be explained in the Stryd vertical GRFs modelling itself where passive peak is missed [55]. Moreover, as discussed by the authors, the relevance of the OptoGait system would not be as accurate as our system of reference (force platform), which could explain divergent findings.

Finally, the LSS measures did not show any major differences between the Stryd power meter and the force platforms (Table 2, Figure 5c). On the one hand, relationship found in the present study between leg stiffness and stride frequency was in agreement with the literature [38]. In addition, Morin et al. [59] found that 90% to 96% of changes in leg stiffness was accounted for by changes in GCT, whereas step frequency indirectly influenced leg stiffness through its relationship with GCT. On the other hand, the leg stiffness represents the lower-limb resistance to deformation and reflects in some way the elastic energy storage and recoil. Thus, leg stiffness has been considered as a kinetic factor related to running economy (RE) by authors [60,61]. RE is recognised to be one of the main determinant of the endurance running performance [62]. Thereby, LSS could be a relevant training parameter in which the runner should major it by mainly shortening the GCT (in addition to resistance and specific training). However, further studies involving changes in GCT, stride frequency and kinematic factors (e.g., angle of attack) for a given speed would assess whether the Stryd power meter is sensitive enough to correctly estimate RE through the LSS.

### 4.3. Other Measures

Our protocol did not allow us to assess the validity of distance and pace. Participants wore their own running watch and GPS signal or auto-calibration was not systematically turned off. By observing the raw data through the FIT files, GPS signal recorded by the watch overwrote estimates of these measures. Consequently, cautions should be taken using the Stryd power meter coupled with a sport watch in practice. We recommend users to turn-off GPS signal and auto-calibration at the expense of somehow useful GPS information (e.g., GPS traces in trail running).

## 5. Conclusions

In this paper, we focused on the Stryd ground contact time, power output and leg spring stiffness validity compared to reference systems measures. The power meter provided acceptable measures of GCT and LSS over the test but interrogations persist about absolute values of PO. Nonetheless, Stryd power meter can be a useful tool to quantify the intensity during sub-maximal running. By correcting absolute PO values of the Stryd power meter, it allows the runner to monitor training loads (e.g., through the external work) and performances across sessions. Recreational runners interested in health rather than performance can also profit of these measures by practising safely. Further analyses remain necessary to assess the power meter validity at higher speeds (maximal and supra-maximal), with direction changes, non linear accelerations and slope.

## Figures and Tables

**Figure 1 sports-08-00103-f001:**
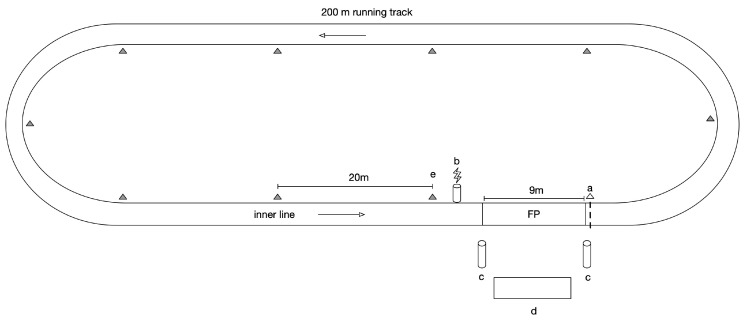
MAS test protocol on a 200 m indoor track. The symbol *a* represents the start line, *b* is a photoelectric cell to reset the force platform records, both *c* are the two motion analysis sensor modules, *d* is the control panel and *e* are the cones laid every 20 m and FP is the force platform recording area.

**Figure 2 sports-08-00103-f002:**
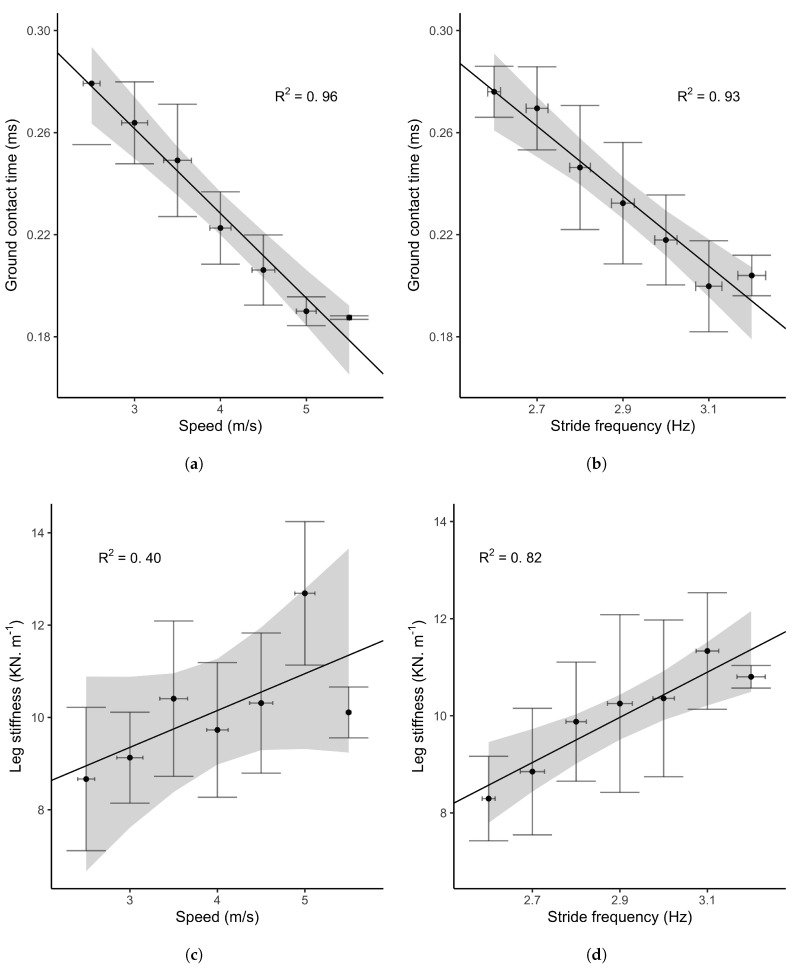
Mechanical stride changes during the MAS test. The top plots (**a**,**b**) represent changes in GCT over speed and stride frequency respectively. The bottom plots (**c**,**d**) represent changes in leg stiffness ($k_{leg}$) over speed and stride frequency according to McMahon and Cheng [22]. In each figures, dots represent the group mean values, error bars the standard deviation in both x, y axes. The solid line is the regression line from the Bayesian linear model, surrounded by the 95% credible intervals.

**Figure 3 sports-08-00103-f003:**
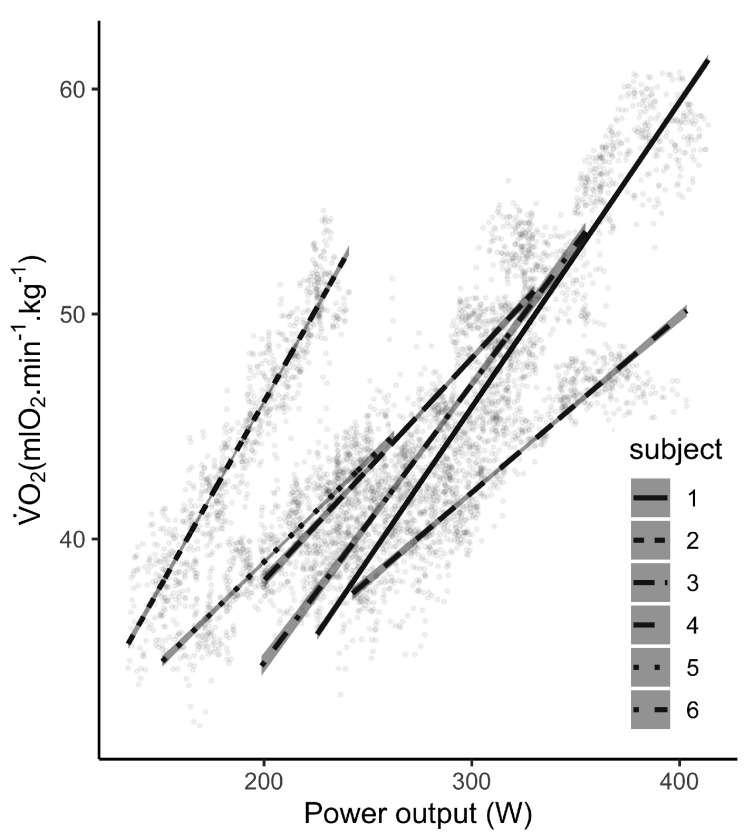
Consumed metabolic energy ($\dot{V}O_{2}$) – Stryd power output (PO) relationship during the incremental test. A strong and positive linear relationship was observed across participants. Lines represent each individual linear regression between $\dot{V}O_{2}$ and PO.

**Figure 4 sports-08-00103-f004:**
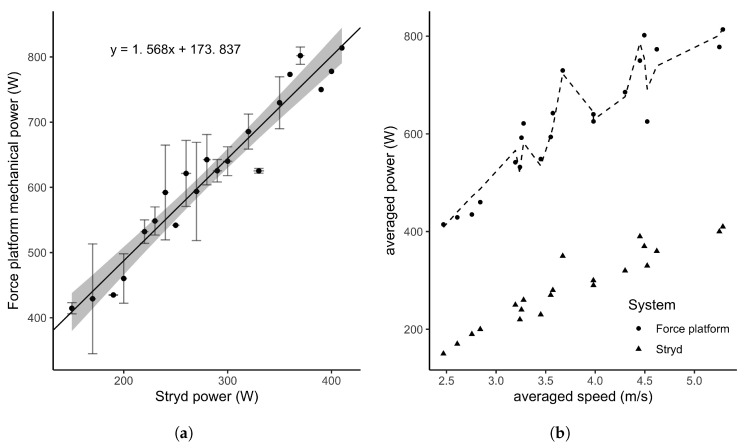
Comparison of PO estimated by the Stryd power meter and the force platform. The left plot (**a**) represents the strong positive relationship between the Stryd and the reference measures. The right plot (**b**) represents the averaged PO in response to speed, where the dotted line is the corrected Stryd PO (see text for details).

**Figure 5 sports-08-00103-f005:**
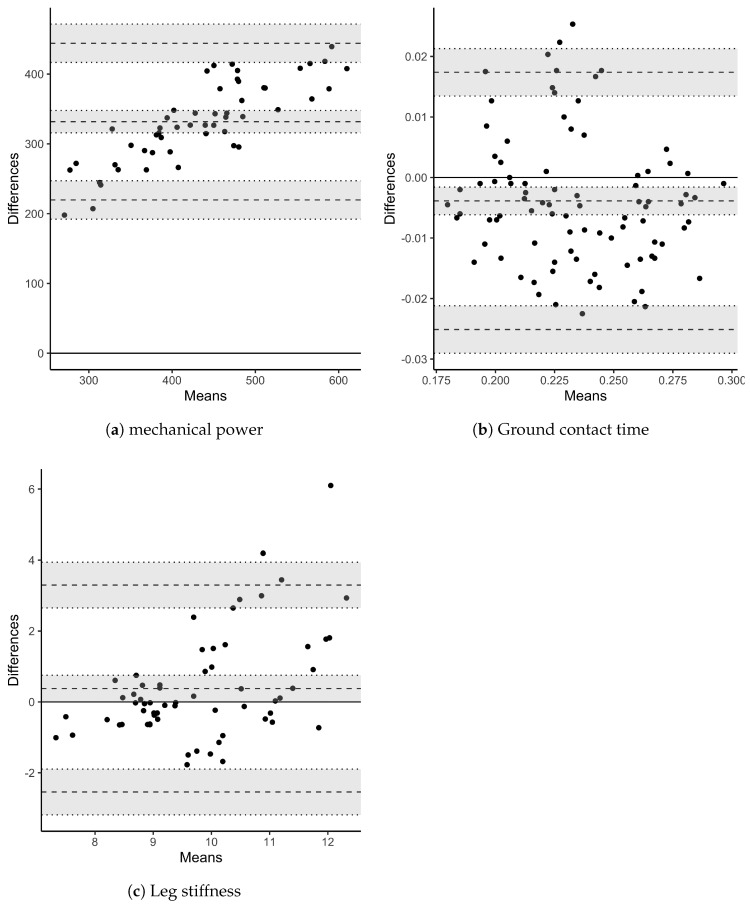
Bland–Altman plots for comparison of measurements between the force platforms (reference) and the power meter. Mean bias (middle dashed line), lower and upper limits of agreement (dashed lines) and their 95% confidence interval areas are represented.

**Table 1 sports-08-00103-t001:** Linear mixed modelling of consumed metabolic energy and mechanical power relationship.

Parameter	Estimate	Est.Error	${\mathrm{CI}}_{\mathrm{lower}}$	${\mathrm{CI}}_{\mathrm{upper}}$	Effects
Intercept	−8.530	8.842	−27.898	7.197	Population-level effects
Mechanical power	0.081	0.018	0.035	0.111	Population-level effects
sd(Intercept)	15.234	7.535	3.123	32.769	Group-level effects
sd(mechanical power)	0.029	0.019	0.004	0.078	Group-level effects
cor(Intercept,mechanical power)	−0.492	0.428	−0.949	0.658	Group-level effects
sigma	2.410	0.346	1.853	3.204	Family specific parameters

**Table 2 sports-08-00103-t002:** Bayesian linear mixed models parameters estimates.

Parameter	Estimate	Est.Error	${\mathrm{CI}}_{\mathrm{lower}}$	${\mathrm{CI}}_{\mathrm{upper}}$	${\mathrm{BF}}_{10}$	Effects	Measure
Intercept	568.401	38.163	490.765	644.830		Population-level effects	Mechanical power
Stryd	−304.952	9.817	−324.304	−285.993	>100	Population-level effects	Mechanical power
Speed	65.486	12.914	42.022	93.125	15.38	Population-level effects	Mechanical power
Stryd:speed interaction	−23.782	9.884	−43.192	−4.554	>100	Population-level effects	Mechanical power
sd(Intercept)	86.027	36.563	41.970	180.565		Group-level effects	Mechanical power
sd(speed)	17.260	18.408	0.438	66.867		Group-level effects	Mechanical power
cor(Intercept,speed)	0.349	0.516	−0.819	0.978		Group-level effects	Mechanical power
sigma	52.325	3.651	45.751	59.970		Family specific parameters	Mechanical power
Intercept	0.241	0.008	0.226	0.255		Population-level effects	Contact time
Stryd	−0.005	0.002	−0.009	−0.002	0.008	Population-level effects	Contact time
Speed	−0.034	0.012	−0.058	−0.011	>100	Population-level effects	Contact time
Stryd:speed interaction	0.000	0.002	−0.003	0.004	0.72	Population-level effects	Contact time
sd(Intercept)	0.016	0.009	0.007	0.039		Group-level effects	Contact time
sd(speed)	0.025	0.015	0.009	0.063		Group-level effects	Contact time
cor(Intercept,speed)	−0.216	0.410	−0.862	0.647		Group-level effects	Contact time
sigma	0.010	0.001	0.009	0.012		Family specific parameters	Contact time
Intercept	8.574	0.980	6.680	10.571		Population-level effects	Leg stiffness
Stryd	−0.602	0.893	−2.334	1.154	0.007	Population-level effects	Leg stiffness
Speed	0.394	0.240	−0.099	0.865	0.012	Population-level effects	Leg stiffness
Stryd:lap interaction	0.063	0.244	−0.418	0.534	0.020	Population-level effects	Leg stiffness
sd(Intercept)	1.427	0.959	0.119	3.830		Group-level effects	Leg stiffness
sd(speed)	0.284	0.255	0.010	0.940		Group-level effects	Leg stiffness
cor(Intercept,speed)	−0.105	0.572	−0.953	0.927		Group-level effects	Leg stiffness
sigma	0.972	0.063	0.857	1.106		Family specific parameters	Leg stiffness

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
