# Peer review of "Validity of the Stryd Power Meter in Measuring Running Parameters at Submaximal Speeds"

_sports, 2020, doi:10.3390/sports8070103_

Round 1
Reviewer 1 Report
The paper details the validity check of a commercial power meter. The reported values are compared with reference measurements provided by an optical motion capture system and force platforms.
Please find the minor comments below.
line 91: What does ''carbon fiber reinforced inertial sensor'' mean?
line 95: It is okay that the Stryd does not need calibration. However, some kind of data should be porvided, such as body height, body weight. Otherwise the power output estimation is not possible. Is it true for the Stryd power meter? Please make it clear in the text.
line 126: Instead of ''capture the vertical, horizontal and rotational movements '' isn't it easier to say ''3D position and orientation is captured''?
line 138: It is not 100% clear whether C1 and C2 are time durations (intervals) or time instances.
2.5.1: Although, GRF is mentioned here, it is not clear whether the GRF was included in the estimation of the GCT. If not, why GRF is mentioned here? Please make it clear.
line 162, equation (1): the dimensions and the symbolic variables are mixing. In line 164, J kg^(-1) m^(-1) should be written with roman characters, since they are dimensions. And only the mathematical variables should be written in italic. Actually, this issue should be corrected everywhere all over the manuscript.
Are there any studies, on which the calculations from line 143 to line 161 are based? There are serious simplifications and neglections. Therefore it would be necessary to have some reference, which proove that these calculations are accurate enough.
line 173: is max a function? If yes, please use max(...).
line 264: Does m on the power of 1 make a sense?
3.1.3: I cannot find the definition of the null hypothesis and the alternative one.
Figure 2: The scaling of the vertical axes of (a) and (b) should be the same. The same for (c) and (d).
Reviewer 2 Report
There are several studies on various parameters including running cost, but there are not so many studies to verify the reliability of the Stryd power meter. The authors tried to verify the reliability of the parameters measured by the Stryd power meter. The present study provides some interesting findings, although the study includes some limitations such as the limited number of subjects and the fact that both sexes are mixed. In my opinion, there seems to be no problem with the research method. I have only a few minor questions and advice.
- Line 68: Which manufacturer is the Stryde power meter?
- Lines 81-82: Although the authors described that the power meter measure is not homogeneous while speed raises (from 8 km/h to approximately 19 km/h) in the lines 336-337, should you specify the final exercise time and speed here? As described in Abstract, it is better to specify it in the text.
- Lines 243-245: It should be specifically explained how “the vague and weakly informative priors” could be modified on the basis of the literatures. By doing so, the reader's understanding of the calculation of 2.6.3 and 2.6.4 will be further improved.
- Line 315: What are the two specific things that "both" indicate? Stlyd power meter and MAS test?
- Line 401: What does "the later" mean?
